# Hypercapnic Failure in Acute Exacerbated COPD Patients: Severe Airflow Limitation as an Early Warning Signal

**DOI:** 10.3390/jcm12010258

**Published:** 2022-12-29

**Authors:** Achim Grünewaldt, Norbert Fritsch, Gernot Rohde

**Affiliations:** 1Department of Pneumology, University Hospital, Goethe University, 60590 Frankfurt, Germany; 2Department of Anesthesiology, University Hospital, Goethe University, 60590 Frankfurt, Germany

**Keywords:** COPD, comorbidity, hypercapnic failure, exacerbation, non-invasive ventilation

## Abstract

Background: Hypercapnic failure is a severe complication of COPD disease progression, which is associated with a high morbidity and mortality. The aim of this study was to examine the association of comorbidity and clinical risk factors with the development of hypercapnia in acute exacerbated COPD patients. Methods: In this retrospective monocentric cohort study, we examined the influence of the clinical parameters and the comorbidity of hospitalized patients with the acute exacerbation of COPD on the development of hypercapnia by performing multivariate logistic regression and a receiver operating characteristic analysis. Results: In total, 275 patient cases with COPD exacerbation were enrolled during the period from January 2011 until March 2015, where 104 patients (37.8%) with hypercapnia were identified. The logistic regression analysis revealed severe airflow limitation (decreased FEV1) as the main factor associated with the development of hypercapnia. In the ROC analysis, we determined an FEV1 of 42.12%, which was predicted with a sensitivity of 82.6% and specificity of 55%, and an absolute value of FEV1 of 0.8 L, with a sensitivity of 0.62 and specificity of 0.79 as the cut off points, respectively. We could not verify an association with the patient’s condition or the laboratory surrogate parameters of organ failure. Conclusion: Severe airflow limitation is an important risk factor that is associated with hypercapnic failure in acute exacerbated COPD patients. Validation in prospective cohorts is warranted and should focus on more intensive monitoring of these at-risk patients.

## 1. Introduction

Acute and chronic hypercapnic failure is a severe complication of chronic obstructive lung disease (COPD) with prognostic relevance regarding mortality and risk of hospitalization [1,2]. 

In the context of a COPD exacerbation, acute hypercapnic failure can be accompanied by respiratory acidosis and is a frequent clinical finding. Roberts and co-workers reported that in up to 20% of patients with COPD exacerbations admitted to hospital, acidotic blood gases were recorded [3]. Hence, the early start of non-invasive ventilation is strongly recommended. Of note, delayed hospital admission of these patients is associated with increased in-hospital mortality [4]. 

The importance of acute COPD exacerbations for increased morbidity and mortality has been extensively shown [5,6]. The main risk factors for acute exacerbations are inadequate therapy and a history of frequent exacerbations in the past [7]. 

A recent analysis by Bekov et al. revealed that a reduced forced expiratory volume in one second (FEV1) was associated with an increased exacerbation rate and mortality [8]. Respiratory viral infections have been identified as important triggers of COPD exacerbations [9]. Furthermore, the current analysis demonstrates decreasing exacerbation rates in COPD patients during the COVID-19-pandemic waves. This is supposed to be a consequence of reduced respiratory viral infections during this period and underscores the significance of viral respiratory tract infections as triggers of COPD-exacerbations [10]. 

In contrast to acute hypercapnic failure, chronic respiratory failure is usually characterized by the absence of respiratory acidosis due to renal compensation by the increased urinary excretion of hydrogen ions and the resorption of HCO3^−^. It has been shown that patients with compensated hypercapnic respiratory failure suffer from a high mortality and healthcare use, with higher PCO_2_ being associated with worse survival [11].

Accordingly, despite the absence of acidosis, chronic hypercapnic failure has been proposed as an indication for long term home non-invasive ventilation support, and a current metanalysis showed decreased mortality in long-term-NIV-treated patients compared with those receiving standard care [12].

Indeed, both forms of hypercapnic failure are associated with increased mortality in COPD. Hypercapnia influences cardiovascular comorbidity and has effects on the musculoskeletal system [13]. Consequently, recognizing potential risk factors leading to hypercapnic failure could reduce mortality in this cohort. 

A large body of literature demonstrates the high burden of comorbidity in COPD patients. Here, the association between COPD exacerbations and the progression of cardiovascular diseases is of special interest [14]. Additionally, particularly in patients with both hypoxemia and hypercapnia, renal complications have to be considered, as correlations between the serum creatinine levels and the severity of COPD, as well as the exacerbation and hospitalization rate have been reported [15]. 

Furthermore, COPD patients show an increased risk for liver disease [16]. Recent data suggest that recurrent rates of hypoxemia might promote cell damage in patients with preexisting liver cell damage [17]. 

Nonetheless, only a limited number of studies have evaluated the direct role of clinical factors and comorbidity in hypercapnic failure. Accordingly, the purpose of this study was to evaluate potential respiratory and non-respiratory factors possibly predicting the development of hypercapnic failure in COPD patients. Identifying such predictors could help to determine the risk profiles and to develop scores for improving the management of high-risk COPD patients.

## 2. Materials and Methods

Study design: retrospective single center cohort study. 

Setting: Department of Pneumology, Center of Medical Care of the University Hospital of Frankfurt/Main, Germany.

The study protocol was endorsed by the local ethics committee (study number 64-15). In consideration of the retrospective design and the use of anonymized data, written informed consent was not necessary.

Patients or the public were not involved in the design, conduct, reporting, or dissemination plans of our research.

### 2.1. Case Definition

The objective of the present study was to assess the association of respiratory (current lung function results) and non-respiratory parameters (sociodemographic data and information about comorbidity, including the results of laboratory routine examinations) with development of hypercapnic failure. The assessed laboratory results encompass the surrogate parameters of renal and liver function (creatinine and aminotransferases). 

Therefore, we collected retrospective data from all hospital admissions due to acute COPD exacerbation (diagnosed by clinical presentation at admission) during the period from January 2011 until March 2015. No patient was included more than once into the study cohort.

We compared the recorded sociodemographic and clinical parameters between the group of patients with hypercapnia (PaCO_2_ > 45 mmHg) and the patients with normocapnia (PaCO_2_ < 45 mmHg) at the time of hospital admission, regardless of medical therapy.

Patients were identified based on ICD (International Statistical Classification of Diseases and Related Health Problems) Code J44. The data were extracted from electronic health records of the hospital data system “AGFA-Orbis”. 

Patients aged < 18 a or with a history of asthma (referring to anamnestic information) were excluded.

### 2.2. Data Analysis 

Data were documented in the software Office Excel 2016. The statistical analysis was performed using the local statistic software “Bias” (version 11.12-05/2020, epsilon) and the statistic software “SPSS” (IBM SPSS Statistics version 27).

The Kolmogorov–Smirnov test was performed for testing of the data distribution. Normally distributed data were described by the mean value and standard deviation. Not normally distributed data were described by the median and range. 

The association between hypercapnia and the determining factors was examined with the univariate analysis. We employed the Wilcoxon–Mann–Whitney test for continuous variables respectively the χ^2^-Test for discrete variables to analyze their influence on development of hypercapnia. All significant parameters were further analyzed by performing multivariate logistic regression and a receiver operating characteristics-analysis (ROC-analysis). An alpha of 0.05 was used as the cut-off for significance.

## 3. Results

In total, 275 patients with acute COPD exacerbation were enrolled during the period from January 2011 until March 2015, where 171 patients (62.2%) with normocapnia and 104 patients (37.8%) with hypercapnia were identified. The median age in both normocapnic and hypercapnic patients was 72 years, with a range of 39–98 years. 

In all cases, lung function results were available. In hypercapnic patients, the median PaCO_2_ level was 52.2 mmHg (45.9–92.3) and in normocapnic patients it was 38.0 mmHg (28.5–45.0). Here, 32 of the normocapnic patients (18.7%) and 14 hypercapnic patients (13.4%, loss of data in two patients) were treated with systemic corticosteroids at the time of hospital admission. Here, 50 patients had atrial fibrillation at hospital admission, 25 in the hypercapnic and 25 in the normocapnic group. Table 1 summarizes baseline characteristics and laboratory results.

### 3.1. Univariate Analysis

There was a trend to more frequent atrial fibrillation in the hypercapnic group (24% vs. 14.6% in the normocapnic group, *p* = 0.071).

The recorded lung function parameters FEV1, FEV1/VC, VC, and RV/TLC were significantly more impaired in patients with hypercapnia (*p* < 0.001).

There was a trend to higher creatinine levels in the normocapnic group (0.93 mg/dL (0.34–12.0) vs. 0.83 mg/dL (0.28–4.3) in the hypercapnic group, respectively (*p* = 0.058). The median hemoglobin level was with 13.5 g/dL, similar in both groups. 

Additionally, GOT was slightly higher in patients with hypercapnia (27 (10–131) vs. 24 (10–120)U/L). This difference was statistically significantly different (*p* = 0.025).

The overall in-hospital-mortality was 15.63%, where 24 (14.03%) of the normocapnic patients and 19 (18.26%) of the hypercapnic patients died during the hospital stay (*p* = 0.35). (See Table 2)

The χ^2^-test showed no statistically significant difference in in-hospital mortality in both groups (*p* = 0.35).

### 3.2. Logistical Regression and ROC-Analysis

Table 3 summarizes the results of the multivariate regression analysis. 

Performing logistical regression and stepwise backward elimination including the significant parameters from the univariate analysis revealed FEV1 (%predicted value) as the only independent parameter significantly associated with hypercapnia (*p* = 0.04525, odds ratio 0.9448; 95%CI 0.8938–0.9988). 

Via ROC analysis, we selected the optimal threshold (cut off-point) of FEV1 for predicting development of hypercapnia. 

A FEV1 of 42.12% was detected as the cut off point with a sensitivity of 82.6% and specificity of 55.0%, and an absolute value of FEV1 of 0.8 L with sensitivity 0.62 and specificity of 0.79, respectively.

The ROC curve of the FEV cut off, in liters, is shown in Figure 1.

## 4. Discussion

This investigation was conducted in order to recognize risk factors for early hypercapnic failure, which significantly impacts the prognosis of COPD patients. We chose clinical variables such as lung function parameters, which reflect the current COPD status. Moreover, we analyzed the surrogate parameters of organ failure and organ disturbance, such as hepatic aminotransferases and creatinine. With respect to earlier investigations demonstrating the prognostic impact of atrial fibrillation (AF) in COPD patients and the more frequent presence of AF in hypercapnic patients, we analyzed the occurrence of this arrhythmia in our cohort. We observed a trend to more frequent atrial fibrillation, in line with the findings of Terzano et al. [18]. 

FEV1 was the only independent parameter significantly associated with the development of hypercapnia. This observation is in accordance with recent data of a cross sectional analysis by Dave et al., who reported lower FEV1 and prior acute hypercapnic failure as the main factors associated with hypercapnia in COPD [19]. 

We found no relevant association with the other clinical or laboratory surrogate parameters of organ failure.

It should be noted that we did not find a significant association between body mass index and the development of hypercapnic failure. The current research suggests that body mass index may not be the appropriate measure to reflect inspiratory muscle mass. Kyle et al., who evaluated the body mass index, fat-free mass (FFM) index, and body fat mass index in this context, reported an underestimation of fat-free mass through the single use of the body mass index [20]. These data were confirmed by Budweiser and co-workers who revealed that malnutrition is inadequately presented by BMI [21]. Additionally, current data suggest that weight loss and not the absolute BMI is a further important risk factor in nutritional status that should be monitored [22].

As far as comorbidities are concerned, atrial arrhythmias in COPD have to be considered, which are supposed to be a consequence of pulmonary hypertension and increased intrathoracic pressure [14,23]. 

In our study, we found a trend to a higher prevalence of atrial fibrillation in the hypercapnic group. It was shown earlier that impaired pulmonary function, hypercapnia, and high values of pulmonary artery systolic pressure are independent predictors of incident atrial fibrillation [18]. Hence, our data support these findings and future research should investagate this further in order to understand the exact pathomechanisms.

In addition, renal and hepatic comorbidity is frequent in COPD patients. In a recent analysis, Yong et al. demonstrated that hepatic fibrosis, assessed using the fibrosis-4 index (FIB-4), is a prognostic factor for COPD mortality and acute exacerbations. In this study, the fibrotic index was assessed using aminotransferases, age, and platelet count [24]. In our cohort, we did not find an association between the assessed surrogate parameters for the liver and hypercapnia. However, this may be due to the different study design and sample size.

Data about renal disease as a prognostic factor in COPD patients are scarce. Ucgun et al. have shown that creatinine is an independent factor for increased mortality in COPD patients with respiratory acidosis [25]. In our cohort, there was no significant association between creatinine and hypercapnia. This could be explained by the fact that creatinine is not a very sensitive marker for the early detection of renal disease. Further investigations using more sensitive markers of renal function are necessary to examine renal involvement in lung disease. 

The interactions between comorbidities, COPD in general, and hypercapnia in particular are still a matter of intensive research and debate. Greulich et al. demonstrated that airflow limitation in COPD seems to not be influenced by comorbidities [26]. Indeed, our data suggest that the leading cause for the development of hypercapnia is the progressive respiratory disease itself. This is comprehensible, bearing in mind that progressive emphysematic remodeling is associated with increased air trapping, impaired diaphragmatic function, and a higher respiratory muscle load. Indeed, exercise testing has shown that hyperinflation contributes to elevated PCO_2_ levels [27].

## 5. Limitations

Finally, the study is subject to the inherent limitations of any retrospective-based data. Retrospective data lack a systematic assessment of comorbidity, which might influence the results, particularly the distribution of comorbidities. We aimed to overcome this by basing our analyses mainly on the measurement of objective parameters such as laboratory values and ECG. A suitable score for planning future studies might be the Charlson index [28]. Sleep apnea as a potential cofactor for hypercapnia was not explicitly assessed. This should be included in future studies.

Concerning the results of the lung function tests, it has to be considered that most patients received the examination during the acute exacerbation. Further studies have to confirmed our results by prospective evaluation of the lung function parameters also during stable disease. 

Another shortcoming is the absence of a standardized risk assessment using established ICU-assessments such as the SOFA or Apache score. Additionally, in our cohort, no follow up examinations were possible due to the study design. 

## 6. Conclusions

The most important finding of our study is that the leading cause for the development of hypercapnia is the progressive respiratory disease of COPD itself. Even though comorbidities in COPD patients occur frequently, we did not find an association with surrogate parameters of renal or liver function and hypercapnic failure in our cohort. We found a trend to more frequent atrial fibrillation in the hypercapnic group, as reported earlier. It is unclear whether this represents a consequence or a possible cause. In general, the role of cardiovascular, renal, or hepatic comorbidity in the development of hypercapnia in COPD patients is not fully understood. In this context, initial experimental studies reporting disturbed alveolar epithelial cell function and alternations of the immune response in the presence of hypercapnia provide a basis for a better understanding of these complex interactions [29,30].

In conclusion, regarding the prognostic relevance of hypercapnia, COPD patients with very severe airflow limitation should be monitored closely, with attention to increasing PaCO_2_ levels in the arterial blood gas analysis.

## Figures and Tables

**Figure 1 jcm-12-00258-f001:**
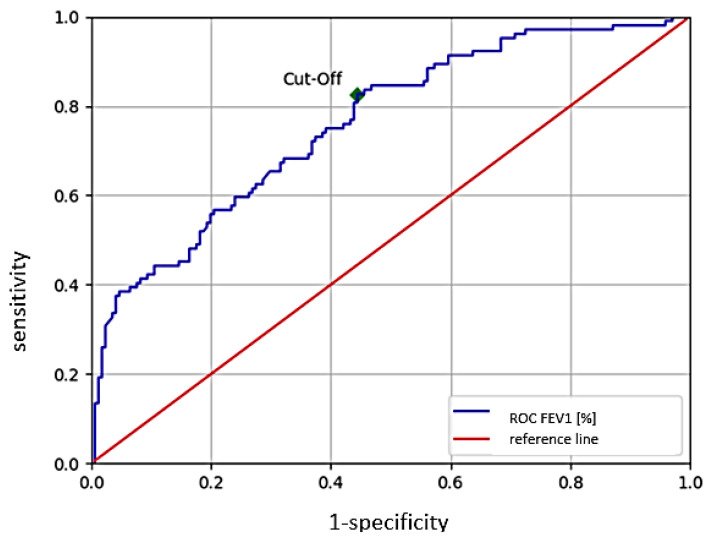
Receiver operating characteristic (ROC) curves of FEV1 [%], with a predicted cut off value of 41.12% with a sensitivity of 82.6% and specificity of 55.0%.

**Table 1 jcm-12-00258-t001:** Baseline characteristics and laboratory results.

	Hypercapnic Patients	Normocapnic Patients	*p*-Value
Age (median), (range 39.0–98.0, Q1 65.0; Q3 77.0)	72	72	-
Male (*n*/%)	62 (59.6%)	101 (58.1%)	-
BMI (range)	24.67 (12.7–60.23)	24.69 (12.7–57.33)	0.896
Packyears (range)	50 (5–250)	45 (1–200)	0.27
Continued smokers (*n*/%)	48 (48.3%)	70 (53.9%)	0.401
Atrial fibrillation (*n*/%)	25 (24.0%)	25 (14.6%)	0.071
OCS at time of admission (*n*/%)	14 (13.4%)	32 (18.7%)	0.369
PaCO_2_ [mmHg] (range)	52.2 (45.9–92.3)	38.0 (28.5–45.0)	-
FEV1 [l] (range)	0.74 (0.34–1.97)	1.10 (0.4–3.6)	<0.001 *
FEV1 [% norm] (range)	29.61 (12.28–84.9)	43.87 (11.65–150.0)	<0.001 *
FEV1/FVC (range)	40.9 (12.8–73.58)	48.03 (15.03–98.09)	<0.001 *
VC [l] (range)	1.63 (0.31–3.77)	2.30 (0.58–5.57)	<0.001 *
RV/TLC (range)	69.57 (28.74–89.71)	60.05 (30.11–83.61)	<0.001 *
Creatinine [0.5–0.9 mg/dL] (range)	0.83 (0.28–4.3)	0.93 (0.34–12.0)	0.058
Hemoglobin [11.6–15.5 g/dL] (range)	13.5 (8.6–19.0)	13.5 (7.1–18.6)	0.595
GOT [<35 U/L] (range)	27 (10–131)	24 (10–120)	0.025 *
GPT [<35 U/L] (range)	22 (4–176)	21 (2–138)	0.434

BMI = body mass index; OCS = oral corticosteroids; FEV1 = forced expiratory volume in 1 s; VC = vital capacity; RV = residual volume; GOT = glutamic oxaloacetic transaminase; GPT = glutamic pyruvic transaminase, expressed in median and range, p-value of univariate analysis with determining factors and hypercapnia; blood sample results with normal reference, * statistical significance alpha = 0.05.

**Table 2 jcm-12-00258-t002:** In-hospital mortality in normocapnic and hypercapnic patients.

	in Hospital-Mortality			*p*-Value (Chi-Square-Test)
	Alive (n)	Deceased (n)	All Patients (n)	
Normocapnic patients	147 (85.96%)	24 (14.03%)	171	
Hypercapnic patients	85 (81.73%)	19 (18.36%)	104	
All patients (n)	232 (84.36%)	43 (15.63%)	275	0.35

**Table 3 jcm-12-00258-t003:** Risk factors in multivariate regression analysis (significant in univariate analysis).

Variable	Beta	SD (Beta)	*p*-Value	Odds-Ratio	95%-CI	
FEV1 [L]	0.6599	1.3199	0.61712	1.9345	0.1456	25.7066
FEV1 [%]	−0.0567	0.0283	0.04525	0.9448	0.8938	0.9988
FEV1/FVC	−0.0153	0.0207	0.45825	0.9848	0.9456	1.0255
VC [L]	−1.1113	0.5013	0.02663	0.3291	0.1232	0.8792
RV/TLC	−0.0117	0.0282	0.67844	0.9884	0.9352	1.0446
Creatinine [mg/dL]	−0.0112	0.1989	0.95519	0.9889	0.6696	1.4605
GOT [U/L]	0.0127	0.0150	0.39785	1.0128	0.9834	1.0430
Atrial fibrillation	0.3516	0.5361	0.51191	1.4213	0.4970	4.0643

CI = confidence interval; FEV1 = forced expiratory volume; VC = vital capacity; GOT = glutamic oxaloacetic transaminase.

## Data Availability

The datasets used and analyzed during the current study are available from the corresponding author on reasonable request.

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
