# Peer review of "Hypercapnic Failure in Acute Exacerbated COPD Patients: Severe Airflow Limitation as an Early Warning Signal"

_jcm, 2022, doi:10.3390/jcm12010258_

Round 1

Reviewer 1 Report

The author explored significance of comorbidity and clinical manifestation on hypercapnia in acute exacerbated COPD. As a result, the present study showed that airflow limitation could be a parameter which is associated with hypercapnia risk in the pathophysiology. This study is interesting, however, there are several major concerns with this work as outlined below.

(Major comments)

1.  The evidence that severe airflow limitation in COPD could be a critical risk factor of hypercapnia in acute exacerbated COPD in the current study is not surprising. Because levels of airflow limitation are definition of severity of COPD. In short, impact of the current result in field of COPD investigation seemed to be slightly little.

2.  The author should combine table 1 or table 2 with table 3 in order to simplify the present result, because univariate analysis in table 3 is obtained by analyzing data of table 1 and table2

3.  The author should describe the result of logistical regression as a new table because the readers want to know which clinical parameters are selected for variables of multivariate analysis.

4.  Is ROC-analysis in figure 1 the result of not absolute value of FEV1 but predicted value of FEV1? If it is so, the author should correct this figure.

5.  The author should select charlson index as comorbidity risk in order to explore influence of comorbidity on hypercapnia in acute exacerbated COPD. If it is impossible to get this index in the current study, the author should add this point as a limitation.

(Minor comments)

1.  The author should add the result in from line 127 to line 130 as a new table.

2.  Is the word “bei” in line 176 correct? If it is wrong, the author should correct this point.

Author Response

Major issues:

1.The evidence that severe airflow limitation in COPD could be a critical risk factor of hypercapnia in acute exacerbated COPD in the current study is not surprising. Because levels of airflow limitation are definition of severity of COPD. In short, impact of the current result in field of COPD investigation seemed to be slightly little.

We agree with the reviewer, that the single result of airflow limitation as risk factor for hypercapnia is not unexpected. However, our results demonstrated, in which extensity reduced forced expiratory volume influences the development of hypercapnia. Moreover, we examined further potential factors which might contribute to hypercapnic failure. Indeed, we could not prove this association in respect to factors like cardiac arrhythmia, elevated liver enzymes or creatinine or body mass index. Therefore, we discussed further aspects of occurrence of comorbidity which should be evaluated in future.

  1. The author should combine table 1 or table 2 with table 3in order to simplify the present result, because univariate analysis in table 3 is obtained by analyzing data of table 1and table2

Agreed and changed. We summarized the tables and added to the clinical characteristics the results of univariate analysis (new table 1).

  1. The author should describe the result of logistical regression as a new table because the readers want to know which clinical parameters are selected for variables of multivariate analysis.

We absolutely agree that these results must be presented. Consequently, we added the revised table 3.

  1. Is ROC-analysis in figure 1 the result of not absolute value of FEV1 but predicted value of FEV1? If it is so, the author should correct this figure.

Agreed and changed (figure 1)

  1. The author should select charlson index as comorbidity risk in order to explore influence of comorbidity on hypercapnia in acute exacerbated COPD. If it is impossible to get this index in the current study, the author should add this point as a limitation.

We agree with the reviewer, that it would appreciable to assess comorbidity by a validated score. Our retrospective data are not suitable for this. Therefore, we mentioned this shortcoming in the “limitations”.

Minor issues:

  1. The author should add the result in from line 127 to line130 as a new table.

Agreed and changed. We introduced table 2.

  1. Is the word “bei” in line 176 correct? If it is wrong, the author should correct this point.

Changed (line 194).

Reviewer 2 Report

This is a well written and easy to read article. However, some questions arise after reading the work.

First of all, the sample draws attention because the data began to be collected more than 10 years ago. Second, it appears to be a small sample, despite including patients over a period of more than 4 years.

In what percentage of patients was spirometry available?

Was the use of mechanical ventilation considered in patients with hypercapnia? It would be interesting to know the influence of this treatment on the evolution of patients.

It would make the data easier to understand by including the data from the univariate analysis in Tables 1 and 2, and deleting Table 3.

It is advisable to update the bibliography of the "Discussion" section: 62% of the citations are more than 5 years old

Author Response

First of all, the sample draws attention because the data began to be collected more than 10 years ago. Second, it appears to be a small sample, despite including patients over a period of more than 4 years.

We agree, that more current data might be preferable in respect to therapy-statements, having in mind that therapeutic management has been modified in the last decade. However, we did not focus on therapy aspects but association of comorbidity or clinical factors with hypercapnia. For this, we do not suppose that our data are not feasible and that the age influences the study results.
We are aware, that a multicentric evaluation with a bigger sample would increase the validity of our statement. Our retrospective monocentric design explains the sample size.

In what percentage of patients was spirometry available?

In all included cases, lung function results were available. We added this information in “results” line 110.

Was the use of mechanical ventilation considered in patients with hypercapnia? It would be interesting to know the influence of this treatment on the evolution of patients.

All patients were treated with clinical standard of care. For this, mechanical ventilation (especially recommendation for noninvasive home ventilation) were proceeded. We did not focus on hypercapnic treatment and follow up. Regrettably, our data give not enough information to discuss influence of mechanical ventilation. We mentioned in “limitations,” that we are aware, that follow up data are necessary for future examinations.

It would make the data easier to understand by including the data from the univariate analysis in Tables 1 and 2, and deleting table 3.

Agreed and changed in accordance to reviewer 1.

It is advisable to update the bibliography of the "Discussion" section: 62% of the citations are more than 5 years old

Agreed and changed, we revised the bibliography. If suitable, we replaced or supplemented the references by more current studies. To make this obvious, supplemented references were listed separately (page 10).

Reviewer 3 Report

I appreciate the opportunity to review this work. I have two overall comments.

1.- I believe the idea to try to identify patients that will present with hypercapnia when exacerbated is good. it might help us define a population at risk and be more pro-active with treatment. I understand that other the lung parameters the authors were unable to find any other associations with hypercapnia. to make the article more interesting i encourage the authors to expand on this point in the discussion and come with ideas for next step trials as well as potential benefits of findings this characteristics.

2.- even though both groups look similar. I will suggest if possible to stratify the hypercapnic and normocapnic group by BODE score or GOLD (ABCD) recently change now. it will help  the readers to clearly see the groups were equal and the other particular difference was the hypercapnia. if the groups are not equal other factors including on the above metnion scores could also influence the results. 

Author Response

1.- I believe the idea to try to identify patients that will present with hypercapnia when exacerbated is good. it might help us define a population at risk and be more pro-active with treatment. I understand that other the lung parameters the authors were unable to find any other associations with hypercapnia. to make the article more interesting I encourage the authors to expand on this point in the discussion and come with ideas for next step trials as well as potential benefits of findings these characteristics.

Agreed and supplemented. We tried to emphasize that, that mechanisms of deranged blood gases and its association with multi organ damage are not fully understand and should be more evaluated. (See conclusion line 270 ff.)

2.- even though both groups look similar. I will suggest if possible, to stratify the hypercapnic and normocapnic group by BODE score or GOLD (ABCD) recently change now. it will help the readers to clearly see the groups were equal and the other particular difference was the hypercapnia. if the groups are not equal other factors including on the above mention scores could also influence the results.

We approve that the mentioned scores could help to stratify both groups. In the past, in our department there was no standardized assessment of mMRC-, CAT-score and exercise capacity. Furthermore, due to the retrospective design, there are no reliable information about exacerbation rate. Accordingly, it is impossible to stratify our cohort with BODE-score or GOLD-classification.

Round 2

Reviewer 1 Report

Several studies (#1,2 etc) has provided that there are close relationships between COPD exacerbation and airflow limitation. Therefore, the current study has not so much valuable clinical implication for this area even if the author mentioned further potential factors is examined.

#1. Int J Chron Obstruct Pulmon Dis. 2018; 13: 1683- 1690.

#2. BMJ Open. 2014; 4(12):e006171.

Author Response

Several studies (#1,2 etc) has provided that there are close relationships between COPD exacerbation and airflow limitation. Therefore, the current study has not so much valuable clinical implication for this area even if the author mentioned further potential factors is examined.

#1. Int J Chron Obstruct Pulmon Dis. 2018; 13: 1683- 1690.

#2. BMJ Open. 2014; 4(12):e006171.

Thank you for your comment. Please respect, that our study focusses on association between hypercapnia and clinical respectively lung functional parameters. We did not intend to demonstrate an association between airflow limitation and risk of COPD-exacerbation.

At study 1 [Int J Chron Obstruct Pulmon Dis. 2018; 13: 1683- 1690.] : the focus of Kooehorst et al. was to demonstrate association between therapy adherence and mortality in COPD-patients. This study did not demonstrate any association with development of hypercapnic failure.

At study 2 [BMJ Open. 2014; 4(12):e006171]: according to Koehorst et al., this study group evaluated risk factors of exacerbation without publishing any data about hypercapnic failure.

In summarize we suppose, that our data with revealing low airflow limitation including the results auf ROC-analysis are helpful for further studies which focus on development of hypercapnic failure. Notwithstanding the negative results of the examined further factors as elevated liver enzymes or creatine, these results demonstrate, that we need further approaches to understand multi organ involvement in COPD-patients, in particular  in case of hypercapnic failure.

Reviewer 2 Report

Table 1 lacks data: the standard deviation in "age", the "p" value in age and sex...

The current format of table 2 is totally incomprehensible.

There seems to be redundant information in the conclusions.

I would like a complete list of references, rather than the changes separately

Author Response

Table 1 lacks data: the standard deviation in "age", the "p" value in age and sex.

Standard deviation in “age” is inserted.
According to the study protocol, age and gender were not defined as determining factors. For this, univariate analysis was only performed with the clinical parameters which were listed as determining factors in the study protocol (therefore age and sex were excluded).
We clarified this in “statistics” and the legend of table 1.

The current format of table 2 is totally incomprehensible.

Agreed and changed.

There seems to be redundant information in the conclusions.

We tried to strengthen this part.

I would like a complete list of references, rather than the changes separately.

Agreed and changed.
